# *Enterococcus faecalis* Countermeasures Defeat a Virulent *Picovirinae* Bacteriophage

**DOI:** 10.3390/v11010048

**Published:** 2019-01-10

**Authors:** Julien Lossouarn, Arnaud Briet, Elisabeth Moncaut, Sylviane Furlan, Astrid Bouteau, Olivier Son, Magali Leroy, Michael S. DuBow, François Lecointe, Pascale Serror, Marie-Agnès Petit

**Affiliations:** 1Micalis Institute, INRA, AgroParisTech, Université Paris-Saclay, 78350 Jouy-en-Josas, France; Arnaud.Briet@ANSES.FR (Ar.B.); elisabeth.moncaut@inra.fr (E.M.); sylviane.furlan@inra.fr (S.F.); astridbouteau@gmail.com (As.B.); olivier.son@inra.fr (O.S.); françois.lecointe@inra.fr (F.L.); pascale.serror@inra.fr (P.S.); marie-agnes.petit@inra.fr (M.-A.P.); 2Institute for Integrative Biology of the Cell (I2BC), CEA, CNRS, Université Paris-Sud, Université Paris-Saclay, 91198 Gif-sur-Yvette, France; magali.o.leroy@gmail.com (M.L.); micheal.dubow@igmors.u-psud.fr (M.S.D.)

**Keywords:** abortive infection, prophage, adsorption, *Enterococcus*, rhamnopolysaccharide

## Abstract

*Enterococcus faecalis* is an opportunistic pathogen that has emerged as a major cause of nosocomial infections worldwide. Many clinical strains are indeed resistant to last resort antibiotics and there is consequently a reawakening of interest in exploiting virulent phages to combat them. However, little is still known about phage receptors and phage resistance mechanisms in enterococci. We made use of a prophageless derivative of the well-known clinical strain *E. faecalis* V583 to isolate a virulent phage belonging to the *Picovirinae* subfamily and to the P68 genus that we named Idefix. Interestingly, most isolates of *E. faecalis* tested—including V583—were resistant to this phage and we investigated more deeply into phage resistance mechanisms. We found that *E. faecalis* V583 prophage 6 was particularly efficient in resisting Idefix infection thanks to a new abortive infection (Abi) mechanism, which we designated Abiα. It corresponded to the Pfam domain family with unknown function DUF4393 and conferred a typical Abi phenotype by causing a premature lysis of infected *E. faecalis*. The *abiα* gene is widespread among prophages of enterococci and other Gram-positive bacteria. Furthermore, we identified two genes involved in the synthesis of the side chains of the surface rhamnopolysaccharide that are important for Idefix adsorption. Interestingly, mutants in these genes arose at a frequency of ~10^−4^ resistant mutants per generation, conferring a supplemental bacterial line of defense against Idefix.

## 1. Introduction

Enterococci are ubiquitous Gram-positive facultative anaerobic bacteria that colonize the mammalian gastrointestinal tract [1]. In particular, the two species *Enterococcus faecalis* and *Enterococcus faecium* are part of the normal human gut microbiota and generally have no adverse effects on healthy individuals. However, they also represent opportunistic pathogens that have emerged as a leading source of nosocomial infections, particularly in immunocompromised patients [2]. *E. faecalis* and *E. faecium* mostly cause urinary tract infections, peritonitis, bacteraemia, and endocarditis [2]. The clinical importance of these bacterial species is directly related to their antibiotic resistance. The rapid spread of clinical isolates resistant to last resort antibiotics such as vancomycin and daptomycin has been of particular concern and associated hospital acquired infections have become a growing problem [3].

Virulent bacteriophages, i.e., viruses that infect and obligatorily lyse bacteria, have long held promise to treat bacterial infections and combat multi-drug resistant (MDR) bacteria [4]. In the recent years, the complete genome and phage cycle characteristics of a dozen virulent *E. faecalis* and *E. faecium* phages have been reported, emphasizing the growing interest for phage therapy in the future [5,6,7,8,9,10,11,12,13,14,15]. However, similar to the phenomenon of antibiotic resistance, bacterial resistance to phage is also taking place. For instance, mutations in the cell wall protein PIP_EF_ were recently found to provide resistance to two siphophages by limiting phage DNA entry in *E. faecalis* [16]. Apart from this, the modes of defence of enterococci against phages largely remain a *terra incognita*.

Keeping in mind that, in most ecosystems, bacteria co evolve with a plethora of bacteriophages, which imperatively depend on them for their reproduction, it is no surprise that bacteria invest efforts in fighting against them. They deploy for this goal different functions that have been studied for decades (reviewed in [17,18]), and still led to new discoveries [19,20,21,22,23]. The *E. faecalis* species, however, seems to have few acquired and innate immune systems. The only common clustered regularly interspaced short palindromic repeat (CRISPR)-CRISPR associated (Cas) locus found in *E. faecalis* isolates has lost its *cas* genes, and two complete type II CRISPR-Cas systems occur variably across the species [24,25,26]. Furthermore, no restriction-modification (R-M) system is commonly found within the species, and to date only a single type II R-M system has been described in three *E*. *faecalis* chromosomes [27,28]. Finally, no complete defense island system associated with restriction-modification was detected in *E*. *faecalis* to date [22]. The relative scarcity of R-M and CRISPR-Cas systems in MDR strains may facilitate plasmid-encoded antibiotic resistance genes acquisition [29,30].

Although less studied, phages themselves compete for their hosts on the bacterial battlefield [31]. Phage genes involved in anti-phage mechanisms are mainly found in temperate or defective prophages, rather than in virulent phages. During the prophage state (where the temperate phage is stably associated with its host), in addition to repressor-dependent immunity against similar phages, some phage express genes conferring resistance to infection by more or less unrelated phages. These genes can encode generalist functions such as R-M systems [32] or CRISPR-Cas loci [33], or specific ones. For instance, the *Escherichia coli* temperate phage P2 has three genes, *fun*, *tin*, and *old*, preventing growth of T5, T-even, and lambda respectively [34]. Furthermore, the O-antigen acetylase of *Salmonella typhimurium* phage BTP1 is reported to prevent adsorption of other phages [35]. Other typical prophage-encoded superinfection exclusion (Sie) systems act to block phage DNA injection into *E. coli* [36], *Lactococcus lactis* [37], and *Streptococcus thermophilus* [38]. Similar systems have also been recently discovered in *Pseudomonas* [39] and *Mycobacterium* prophages [40].

Prophages can also encode abortive infection (Abi) mechanisms that cause an interruption of invasive phage development and a premature death of the infected bacteria. This leads to the release of few or no progeny particles and thus prevents the expansion of the infection to the neighboring bacteria [17,18]. Abi are very diverse; among those carried by prophages, some have been well characterized such as the two component Rex system preventing lambda-lysogenic *E. coli* strain infection by T4 phage [41] or the tyrosine kinase Stk of coliphage 933W that blocks the replication cycle of HK97 [41,42].

Our earlier work showed that the vancomycin-resistant *E. faecalis* V583 clinical isolate hosts seven prophage elements. One prophage is remnant and completely domesticated (prophage 2) while the six others have various degrees of autonomy and different levels of interference with each other [43]. Only three of them—prophages 1, 3, and 5—are fully active and grow as plaques on a V583 derivative cured of all six prophages. However, prophage 1 is parasitized during the induction of it lytic cycle by the satellite prophage 7 also named a phage inducible chromosomal island. Prophages 4 and 6 finally retain some phage-like behavior but no longer produce infective virions. They excise from the bacterial chromosome in a process controlled by the other prophages, and once excised, prophage 4 replicates, while prophage 6 cannot [43].

To investigate putative anti-phage roles played by V583 prophages, we performed a phage screening using the V583 derivative strain cured of all plasmids and the six active prophages. We isolated a virulent phage belonging to the *Picovirinae* subfamily and to the P68 genus that contains phages with small size genome (19–20 kb). This prompted us to name this phage Idefix, after the French name of the small dog from Asterix comics. The genome sequence of two phages similar to Idefix have been reported recently [13,44]. We showed that *E. faecalis* prophage 6, which apparently belongs to the *Siphoviridae* family, is especially efficient in resisting to Idefix infection due to a new abortive infection system that we call Abiα. This latter confers a typical Abi phenotype, by causing a premature lysis of infected *E. faecalis*. The *abiα* gene is notably widespread among prophages integrated in enterococci and other Gram-positive bacteria. Furthermore, the bacterium itself provides another line of defence against Idefix through efficient mutagenesis of the phage bacterial receptor encoded within the variable part of the *epa* locus responsible for the building up of the surface rhamnopolysaccharide of *E. faecalis*. 

## 2. Materials and Methods

### 2.1. Sample for Bacteriophage Isolation, Bacterial Strains, Plasmids, and Growth Conditions

One raw sewage water sample from the Sèvres wastewater treatment plant (Paris area, France) was used as a source of phage. *E. faecalis* indicator strain (cured of active prophages (strain VE18590)) used for phage Idefix isolation, as well as strains and plasmids used for *E. faecalis* phage-resistance characterization are listed in Appendix A (with Appendix A [45,46,47,48,49,50]). *Enterococcus* strains tested to determine phage Idefix host range are detailed in Appendix A (with Appendix A [51,52]). *E. coli* was cultivated at 37 °C in LB medium with shaking. *L. lactis* was grown statically at 30 °C in M17 medium supplemented with glucose 0.5% (M17G). *Enterococcus* strains were cultivated statically at 37 °C either in BHI medium or in M17G medium. Following antibiotics were added when necessary: erythromycin, 10 µg·mL^−1^ for *E. faecalis* and 100 µg·mL^−1^ for *L. lactis* and *E. coli* strains; chloramphenicol, 20 µg·mL^−1^ and 10 µg·mL^−1^ for *E. coli* and *E. faecalis* strains, respectively; kanamycin, 50 µg·mL^−1^ for *E. coli* strains.

### 2.2. Phage Isolation

Phage Idefix was isolated using the standard double overlay plaque assay technique as previously described [44] with minor modifications. A standard Petri dish was filled with 20–30 mL of BHI medium containing agar 1.5% and MgSO_4_ 10 mM. One hundred microliters of 0.45 µm filtered sewage was mixed with 500 µL of an overnight culture of the indicator strain, then added to 5 mL of BHI medium containing agarose 0.4% and MgSO_4_ 10 mM, and poured onto the bottom agar. The double agar/agarose plate was incubated 24 h at 37 °C and screened for plaque appearance. Many different plaques were obtained, and among them, a large clear plaque was picked and streaked on an agar base before applying the second layer of BHI top agarose mixed with the overnight culture of the indicator strain. The plate was incubated 24 h at 37 °C and a large clear plaque was streaked again twice, to ensure phage purity.

### 2.3. Phage Concentration and Purification

Starting from a large clear plaque, virions were resuspended in 1 mL of SM buffer (Tris-HCl 10 mM pH 7.5, MgSO_4_ 10 mM, NaCl 300 mM) and the suspension was centrifuged 10 min at 7500× *g* and room temperature. Phage titer in the supernatant was determined by preparing serial dilutions in SM buffer and using the double overlay method. High-titer phage stock was obtained as previously described [53] with slight modifications. Five hundred to one thousand phages were plated using the double overlay method, which led to a confluent lysis after 12 h at 37 °C. The plate was then flooded with 5 mL of SM buffer and incubated 2 h at 4 °C. The overlay was carefully collected and centrifuged 20 min at 7500× *g* at room temperature and the phages-containing supernatant was passed through a 0.45 µm filter. Phage stocks were titrated (~10^11^ PFU·mL^−1^) and stored at 4 °C for further experiments. Ten milliliters of filtered phage stock was incubated 12 h at 4 °C under stirring in the presence of NaCl 1M and PEG 8000 10 %. Precipitated phages were then centrifuged 20 min at 10,000× *g* and 4 °C and slowly resuspended in 1 mL of SM buffer for 1 h at 4 °C prior to be purified by centrifugation in a CsCl buoyant gradient. The purified phage-containing fraction was then recovered (density 1.4), titrated and stored at 4 °C for further experiments.

### 2.4. Phage Examination in Transmission Electron Microscopy

Ten microliters of purified phage Idefix fraction were directly spotted onto a Formwar carbon coated copper grid. Phages were allowed to adsorb to the carbon layer for 5 min and excess of liquid was removed. Ten microliters of a staining uranyl acetate solution (1%) was then spotted to the grid for 10 s and excess of liquid was removed again. The grid was imaged at 80 kV in a Hitachi HT7700 transmission electron microscope.

### 2.5. Phage Genomic Nucleic Acid Extraction, Whole Genome Sequencing, and Bioinformatic Analysis

Total DNA was extracted as described in [54,55]. Prior to DNA extraction, 10 mL of the phage stock (10^11^ PFU·mL^−1^) was treated with 40 µL of nuclease mix (50% glycerol, 0.25 mg·mL^−1^ RNAse A, 0.25 mg·mL^−1^ Dnase I, 150 mM NaCl), for 15 min at 37 °C. Particles were then precipitated by adding PEG 8000 (10%, *w*/*v*) and NaCl to 1 M, and let sit overnight at 4 °C once PEG was solubilized. Phages were centrifuged 10 min at 10,000× *g* and room temperature, and resuspended into 500 µL of SM buffer. Insoluble particles were removed by centrifugation (20 s at 12,000× *g*), and the clarified supernatant was used for DNA extraction. The PROMEGA Wizard DNA Clean up kit (ref A7280) was then used, following essentially the manufacturer instructions. Prior to DNA elution from the column, a washing step with 5.4 M guanidium thiocyanate (resin solution) was applied. DNA was sent to a 454-sequencing platform, and reads were assembled with Newbler [56]. The phage genome was annotated using RAST [57], followed by manual inspection. The genome sequence is available under the accession number LT630001.1. All genomic figures, including Idefix genome comparison, were generated using Easyfig [58].

### 2.6. Determination of Phage Burst Size

The one-step growth kinetic curve of phage Idefix was measured using a standard method [59] with minor modifications. One milliliter of a log-phase culture of indicator strain was centrifuged 5 min at 10,000× *g* and room temperature, and resuspended in 100 µL of prewarmed BHI medium with MgSO_4_ (10 mM). Phage from the high-titer phage stock was added at a MOI of 0.001 and allowed to adsorb for 5 min at 37 °C. The phage/bacteria mix was centrifuged 2 min at 10,000× *g* and room temperature. The supernatant was titrated to count unadsorbed phage particles, whereas the bacterial pellet was washed in 100 µL of prewarmed BHI medium with MgSO_4_ (10 mM) and recentrifuged. The pellet was suspended and diluted in 10 mL of prewarmed BHI medium with MgSO_4_ (10 mM) and cultured at 37 °C. Samples were taken at regular intervals and plated at the correct dilution for phage titration. A second set of samples from a synchronized 100-fold diluted culture was taken at same intervals and titrated. The values indicate the means and standard deviations of three independent experiments.

### 2.7. Determination of Phage Host Range

Ten microliters of serially diluted high-titer phage stock were spotted (10 µL) on top of agarose overlays containing overnight culture of enterococci, as described above (Appendix A). Plates were incubated at 37 °C and examined for plaque appearance 6, 12, and 24 h after spotting.

### 2.8. Determination of Phage Efficiency of Plaquing

Efficiency of plaquing (EOP) was determined for *E. faecalis* derivative strains (Appendix A). One hundred phages were plated using the double overlay plaque assay technique. EOP were calculated as ((phage titer on tested strain) × (phage titer on the indicator strain)^−1^) × 100. These experiments were independently performed three times and average values are reported with standard deviations.

### 2.9. Phage Adsorption Assay

Adsorption of phage to *E. faecalis* derivative strains were determined as reported previously [60] with minor modifications. One milliliter of log-phase cultures was harvested and resuspended in 100 µL of BHI medium with MgSO_4_ (10 mM), and then phage from the high-titer phage stock was added at a MOI of 0.001. Following incubation for 10 min at 37 °C, the phage/bacteria mixtures were centrifuged 2 min at 10,000× *g* and room temperature. Supernatants were plated at the correct dilution and titrated for phage. Percentages of adsorption were calculated as ((control titer - residual titer) × (control titer)^−1^) × 100. These experiments were independently conducted three times and average values are given with standard deviations.

### 2.10. Determination of Phage Efficiency of Center of Infection

Efficiency of center of infection (ECOI) was determined for *E. faecalis* derivative strains as detailed by [61] with minor modifications. One milliliter of log-phase cultures was harvested and resuspended as previously, and then phage from the high-titer phage stock was added at a MOI of 0.001. Following incubation for 5 min at 37 °C, the phage/bacteria mixtures were centrifuged 1 min at 10,000× *g* and 4 °C, washed twice, diluted, and assayed for infective centers. An *E. faecalis* strain which does not adsorb Idefix was used as control to monitor the effectiveness of phage removal during washing. Percentages of ECOI were calculated as ((number of centers on indicator strain) × (number of centers on tested strain)^−1^) × 100. These experiments were independently performed three times and average values are reported with standard deviations.

### 2.11. Bacterial Survival Assay

*E. faecalis* derivative strains survival was assayed as essentially described in [62] with slight modifications. Briefly, one milliliter of log-phase bacterial cultures was harvested and resuspended as previously described, and then phage from the high-titer phage stock was added at both MOI of 1 and 10. Following incubation for 20 min at 37 °C, bacterial suspensions were plated at the correct dilution on agar plates and surviving bacteria were enumerated as CFU. Percentages of bacterial death were calculated as ((CFU·mL^−1^ in cultures without phage − CFU·mL^−1^ in cultures with phage) × (CFU·mL^−1^ in cultures without phage)^−1^) × 100. These experiments were independently conducted three times and average values are shown with standard deviations.

### 2.12. Lysis Curve Experiments

One milliliter of log-phase bacterial cultures was harvested and resuspended as previously described, and then phage from the high-titer phage stock was added at an MOI of 10. Following incubation for 10 min at 37 °C, the phage/bacteria mixtures were centrifuged 2 min at 10,000× *g* and room temperature. Pellets were resuspended in 200 µL of BHI medium with MgSO_4_ (10 mM) and the suspensions were transferred in wells of a 96-wells plate. Empty wells are filled with BHI medium or uninfected bacterial suspensions to have negative and positive growth control respectively. Bacterial growths were then monitored during 30–50 min at 37 °C using a Tecan plate reader (OD 600 nm, measurements at 2 min intervals after shakings). During the experiment, each sample was prepared and monitored in triplicates. Kinetics shown are representative of three independent experiments.

### 2.13. Luria–Delbrück Fluctuation Tests

An exponentially growing *E. faecalis* strain cured of prophage 6 (strain VE18306) culture supplemented with 10 mM MgSO_4_ was distributed in a 96-wells plate (180 µL, corresponding to ~2 × 10^4^ CFU, in 93 wells), and infected with phage Idefix (20 µL, corresponding to ~2 × 10^9^ PFU, which are dispensed before bacteria, in 90 of the 96 wells). Remaining wells contained either BHI medium or uninfected VE18306 cultures. Following static incubation of the plates for 12 h at 37 °C, bacterial growth was evaluated using a Tecan plate reader, after a 1 min shaking step (OD 600 nm). If mutations occur at random in the bacterial population, the number of mutational events per well follows a Poisson distribution [63], and the proportion of wells in which no mutant resisting to Idefix was present at the time of infection P_0_, is related to h, the expectation of this law, by the formula P_0_ = e^−h^. Knowing the average number of bacteria per well at infection, N, mutation frequency *f* = h/N, so *f* = −lnP_0_/N. These experiments were independently performed three times giving a similar value of *f*.

### 2.14. General Molecular Biology Methods

All PCR reactions to clone or sequence *IDF_13*, *IDF_15*, *ef2833, ef2847, ef2850, ef2169*, and *ef2170* genes were performed in a Mastercycler Eppendorf with Phusion high-fidelity DNA polymerase (NEB) and according to manufacturer’s instructions. PCR products and DNA restriction fragments were purified with QIAquick kits (QIAGEN, Hilden, Germany) when necessary. Electro-transformation of *E. coli*, *L. lactis*, and *E. faecalis* were carried out as previously described [64,65] using a Gene Pulser apparatus (Bio-Rad Laboratories, Inc., Hercules, CA, USA). Transformations of chemically competent *E. coli* ER2566 were carried out by a heat shock procedure. 

### 2.15. Cloning of IDF_13 and IDF_15, Expression and Preparation of Extracts of E. coli, and Spot Assay of the Extracts on the Indicator Strain

PCR amplification of *IDF_13* and *IDF_15* were performed using primer pairs OFL264/OFL265 and OFL266/267 respectively, with Idefix genomic DNA as template. PCR products were purified and digested by *NdeI* and *BamHI* restriction enzymes. Digested products were purified and cloned between the *NdeI* and *BamHI* sites of a linearized pJ411 derivative (Appendix A). JM105 transformants were selected on LB plates supplemented with kanamycin. The resulting plasmids, pAB1and pAB2, contained the corresponding ORF fused in 5’ to a sequence coding for a His6-tag, placed under the control of a T7 promoter. His6-tagged IDF_13 and IDF_15 were produced in *E. coli* ER2566 transformed with pAB1 and pAB2 respectively. Cells were grown in 50 mL of LB supplemented with kanamycin at 37 °C. At OD_600_ = 0.6, production of the phage proteins was induced by addition of IPTG (0.5 mM final concentration) to the culture for 2.5 h. Cells were then harvested by centrifugation at 5200× *g* for 7 min at 4 °C and resuspended in 1 mL of lysis buffer (Tris-HCl 50 mM pH 8, NaCl 150 mM). Cells suspensions were stored at −20 °C until preparation of crude extracts. After thawing, bacteria were lyzed by sonication on ice. The lysates were centrifuged at 4 °C for 20 min at 20,000× *g* and the supernatants were stored at −20 °C. Production and solubility of IDF_13 and IDF_15 were verified by SDS-PAGE. Lysis activity of IDF_13 and IDF_15 was assessed by spotting 5 µL of the supernatants alone or a mix of both fractions on an indicator strain bacterial lawn as previously described.

### 2.16. Construction of E. faecalis VE18306 ef2833-, ef2847- and ef2850-Complemented Strains and E. faecalis VE18590 ef2833- Complemented Strain

Complementations of the prophage 6 genes *ef2833*, *ef2847*, and *ef2850* were done using the pJIM2246 vector (Appendix A). The three genes were amplified including their constitutive promoters and ribosome binding sites from DNA of *E. faecalis* strain containing all prophages (strain VE14089) with primer pairs JL12/JL13, JL10/JL11, and JL8/JL9 respectively (Appendix A). The three purified products were separately cloned into pJIM2246 yielding plasmids pJL1, pJL2, and pJL3 after respective transformation into *E. coli* JM105 and chloramphenicol selection. pJL1, pJL2, and pJL3 were then separately electroporated into *E. faecalis* cured of prophage 6 (strain VE18306). The *ef2833-*, *ef2847-*, and *ef2850*-complemented strains were respectively selected on chloramphenicol. The nucleotide sequences of cloned PCR products were systematically confirmed by sequencing using primer pairs OEF879/1233 (Appendix A). pJL3 and pJIM2246 were also separately electroporated into the indicator strain and selected as previously described.

### 2.17. PCR Amplification in epaX Region

*epaX* region was amplified from total DNA of nine Idefix spontaneous resisting mutants using primer pairs OEF394/OEF397 and OEF527/OEF397 (Appendix A). PCR analysis was extended for the three mutants for which no IS insertion was discovered with larger amplifications downstream and upstream *epaX* region using primer pairs OEF857/OEF527 and OEF885/OEF858; and OEF528/OEF856 and OEF859/OEF823 (Appendix A) respectively.

### 2.18. Construction of E. faecalis VE18306∆epaX Strain

A deletion of *epaX* in the VE18306 background was constructed by double homologous recombination using pVE14283 plasmid (Appendix A). pVE14283 was electroporated into strain VE18306, and the *epaX* deletion (strain VE18393, Appendix A) was selected as described in [49,66].

### 2.19. Construction of E. faecalis VE18393 epaX—Complemented Strain

A complementation of *epaX* was constructed using pVE14297 plasmid (Appendix A). This latter, containing *epaX* under the control of the constitutive promoter P_aphA3_, was electroporated into the VE18306 ∆*epaX* strain (strain VE18393) and the complementation (strain VE18945, Appendix A) was selected as described in [49]. A control strain, harboring pVE14176 vector devoid of *epaX*, was also obtained (strain VE18944, Appendix A). 

## 3. Results

### 3.1. Characterization of the Enterococcus Phage Idefix

#### 3.1.1. Isolation, Morphological Characterization, and Phage–Host Relationship

One municipal sewage water sample from Sèvres (Paris area, Ile de France region) was screened by direct plating without enrichment for phages forming plaques on the indicator strain. This latter, VE18590 (here below pp^−^, complete names of all strains used are listed in Appendix A), corresponds to an *E. faecalis* V583 derivative deleted from its endogenous plasmids and six active prophages [43]. A phage making particularly large and clear plaques was isolated with the double-layer technique. After purification and amplification, transmission electron microscopy revealed a virion with an icosahedral head ~40 nm in diameter and a very short non-contractile tail (Figure 1A). This phage, Idefix, therefore belongs to the *Caudovirales* order and the *Podoviridae* family. A one-step growth kinetic indicated that Idefix has a latent period of 15 min and a burst size ~50 PFUs per infected pp^−^ bacterium (Figure 1B).

#### 3.1.2. Genomic Characterization

The genome of phage Idefix is a double stranded linear DNA, consisting of 18,168 bp with inverted terminal repeats 61 bp long and an average GC content of 33.2%. It is highly homologous to *Enterococcus* phages vB_EfaP_IME195 (95% coverage and 92% nt identity) and vB_Efae230P-4 (75% coverage and 85% identity) (Figure 2). These genomes belong to virulent podophages, which were isolated from sewage samples in China and Poland, respectively [13,44]. A close relative phage, vB_EfaP_IME199, infecting an *E. faecium* strain has also been described [14]. Genbank accession numbers of these genomes are listed in Appendix A.

The Idefix genome encodes 25 ORFs, 12 of which were assigned a function (Figure 2 and Appendix A). The replication module contains a single strand DNA binding protein (SSB, IDF_02) and a DNA polymerase belonging to the B type superfamily (Pol, IDF_07). This polymerase presents all the conserved motifs typical of phage phi29 polymerase, including the specific regions of the “protein priming” subfamily [67,68,69,70,71] (Appendix A).We next searched for the gene coding the terminal protein used to initiate the replication at both ends of *Picovirinae* linear genomes. Such genes are generally located near the polymerase gene but very poorly conserved. Terminal proteins share features like a small size and a high isoelectric point [72,73]. Based on these criteria, *IDF_05* appears the best candidate (Appendix A) but additional investigations are needed to confirm this hypothesis. A gene encoding an encapsidation protein (IDF_06) separates this putative terminal gene and the polymerase gene. The lysis module is composed of a holin (Hol, IDF_16) and two different putative endolysins (Lys): IDF_15 and IDF_13. IDF_15 displays homology with *N*-acetylmuramoyl-l-alanine amidases (Appendix A). To test whether this gene expresses an active endolysin, *IDF_15* was cloned in an expression vector based on a T7 promoter (plasmid pAB2) and expressed in *E. coli* ER2566 (Appendix A). Spotting of a soluble cell extract (containing IDF_15, Appendix A) on a pp- bacterial lawn revealed a lysis zone (Appendix A). This result tends to show that IDF_15 has an endolysin activity. IDF_13 displays homology with PlyCA [74,75,76], a subunit of the PlyC endolysin synthetized by the *Streptococcus* podophage C1 (Appendix A). Indeed *IDF_13*, like *plyCA*, encodes a protein with two catalytic domains. A putative glycosidase domain (homologous to the one of PlyCA) and a putative “CHAP” domain (more divergent compared with the one found in PlyCA) are respectively located in the N and the C terminus of the protein (Appendix A). In contrast to IDF_15, a soluble cell extract of an *E. coli* strain expressing the *IDF_13* (via plasmid pAB1) did not lead to a bacterial lysis (Appendix A). Moreover, a mix of extracts containing IDF_13 and IDF_15 did not produce increased lysis (Appendix A). C1 is the only reported phage whose endolysin PlyC is synthesized from two genes: *plyCA* and *plyCB*. To form the complete enzyme, the catalytic subunit PlyCA and eight subunits PlyCB harboring the cell wall binding domain (CBD) are associated [75,77]. However, we were not able to find an Idefix ORF displaying any significant similarity with PlyCB. We concluded that *IDF_15* seems to encode a canonical monomeric endolysin, and that more experiments are required to determine whether *IDF_13* is just remnant of a former endolysin module, or retains activity combined with another Idefix protein to form a multimeric endolysin. Finally, the structural module includes genes encoding a tail protein (IDF_17), connector proteins (IDF_20 and IDF_21) and a major head protein (MHP, IDF_22) (Appendix A). 

Comparative genomic analyses clearly show that Idefix and the other homologous *Enterococcus* phages belong to the *Picovirinae* subfamily [78] composed of small virulent podophages infecting Gram-positive bacteria and encoding a phi29-like DNA polymerase. The *Picovirinae* subfamily is subdivided into P68, phi29, Cp-1 genera and one unassigned genus (Figure 2). The overall synteny and the MHP similarities [14] both lead to assign Idefix and all other *Enterococcus Picovirinae* to the P68 genus, including one *Streptococcus* and some *Staphylococcus* phages (Figure 2). Within this genus, a hallmark of all *Enterococcus* phages is the presence of two genes coding putative endolysins (IDF_13 and IDF_15 in Idefix). 

#### 3.1.3. Phage Host Range

Plaque assays of Idefix were performed on fifty-nine *Enterococcus* strains, belonging to the *E. faecalis* (47) and to the *E. faecium* (12) species. *E. faecalis* strains had sewage, clinical, commensal, or food origins and represented a range of different clonal complexes and capsule types, whereas *faecium* strains essentially belonged to different sequence types from the most prevalent clinical clonal complex (Appendix A). Idefix did not propagate on any of them, including the wild type strain V583 (wt), from which our indicator sensitive strain pp^−^ is derived (Figure 3A,B). 

### 3.2. Characterization of E. faecalis V583 Resistance to Idefix

#### 3.2.1. Novel Abi System Encoded by V583 Prophage 6

To investigate the mechanism by which *E. faecalis* V583 resists Idefix infection, we made use of our collection of V583 derivatives deleted for either plasmids or prophages [43]. The plasmids hosted by V583 were not responsible for the resistance to Idefix, as the use of the plasmidless V583 derivative strain VE14089 (pp^+^, Appendix A), did not permit Idefix growth (Figure 3C). Plaque assays were next performed on V583 plasmidless derivatives where only one of the prophages remains. Idefix grew on all derivatives (Figure 3D–H) but one, VE18581 (pp6^+^) in which prophage 6 is present (Figure 3I). We concluded that prophage 6 is necessary to confer resistance to Idefix. Plaque assays on the V583 derivative VE18306 (pp6^−^) in which only prophage 6 is deleted, led to clear plaque appearance (Figure 3J) with efficiency of plaquing (EOP) similar to pp^−^ (Table 1). We concluded that prophage 6, which structural genes are notably related to the siphophage HK97, is sufficient for Idefix resistance. Interestingly, we noticed that plaque size was reduced (~three times smaller) whenever prophage 3 was present (Figure 3G,H,J) (Table 1), suggesting that prophage 3 slightly interferes during Idefix infection.

To characterize the resistance conferred by prophage 6, adsorption assays were first performed on a panel of strains differing for the presence or absence of this prophage. Idefix adsorption was always efficient, regardless of the presence of prophage 6 (Table 2), showing therefore that prophage 6 does not prevent Idefix adsorption. We next assayed efficiency of center of infection (ECOI) by Idefix on the same panel of strains, which can detect infective events, regardless of phage burst size. It was reduced to 39% upon infection of strain pp6^+^. This reduction was even more drastic when the strain pp^+^, carrying all prophages, was tested, with only 15% of the infections leading to phage production. This suggests again that some other prophage gene(s) impede Idefix growth. However, infection of strain pp6^−^ permitted to recover plaques for nearly 70% of all infecting particles, suggesting that prophage 6 contribution to interference is major (Table 2). We finally tested whether prophage 6 affected cell survival upon Idefix infection. Cell survival following exposure to Idefix was similar irrespective of the presence of prophage 6: even in the Idefix resistant strains, percentage of cell death was around 50–70% at MOI of 1, and around 100% at MOI of 10, like in Idefix sensitive strains (Table 2). We concluded that prophage 6 encodes a typical Abi mechanism affecting the production of virions while leading simultaneously to host cell death upon Idefix infection. 

To search for prophage 6 candidate genes involved in this Abi mechanism, we analyzed transcriptomic data of strain pp^+^ [48]. Eight of the prophage 6 encoded genes are expressed constitutively during normal growth conditions, in contrast to the generally low expression level of the remaining 50 genes carried by the prophage (Figure 4A). Two of these genes, *ef2855* and *ef2852*, encode the integrase and repressor proteins, which are implicated in the lysogenic control of the prophage. Between these two genes, and probably forming an operon with them, genes *ef2854* and *ef2853* encode a putative membrane protein and a putative metallo-peptidase, respectively. A similar gene pair, placed between an integrase and a repressor, is found in several Sie prophage-encoded phage resistance systems within the *Lactococcus* genus [37,79]. These systems block the phage DNA injection step, due to the membrane protein. The nearby metallo-peptidase is dispensable, but sometimes enhances the resistance efficiency [37,79]. Given that Sie systems do not affect bacterial survival, we did not consider the gene pair *ef2853* and *ef2854* as responsible for the observed resistance to Idefix infection. Among the four last candidate genes, *ef2801* is disrupted by a transposable element invalidating a putative glycosyltransferase, so that our investigation was restricted to the three remaining genes *ef2833*, *ef2847*, and *ef2850*.

All of them are preceded by a promoter region including an experimentally mapped transcription-starting site [80] and encode proteins with unknown function. We thus cloned these three genes individually with their own promoters on pJIM2246 vector (Appendix A), and performed Idefix infections in pp6^−^ transformed by each of the plasmids. Plaque assays revealed that Idefix was able to infect all strains (Figure 4B(a,b) but the one hosting plasmid pJL3, in which *ef2833* is expressed (Figure 4B(c)). We additionally tested plasmid pJL3 in the pp^−^ background and found that this strain was also resistant to Idefix (Figure 4B(d)) whereas the phage grew on the isogenic strain hosting the empty vector (Figure 4B(e)) as well as on pp^−^ (Figure 4B(f)). 

We then checked the other proliferation parameters of Idefix on the two strains hosting the pJL3 plasmid. The phage adsorbed as efficiently on these strains, its ECOI was reduced to ~10% and the percentage of cell death were around 60–70% at MOI 1 and around 100% at MOI 10 in both cases (Table 2). These results confirmed that *ef2833* is responsible for the abortive mechanism. As letters from *abiA* to *abiZ* have been used once to designate over 20 *L. lactis* Abi systems [18,81], we renamed *ef2833* as “*abiα*”. The corresponding encoded protein contains 272 residues, and does not exhibit similarity with any domain or protein of known function. Abiα nevertheless constitutes a PFAM domain referred to as DUF4393 (158 sequences, 140 species, both Gram^+^ and Gram^−^), suggesting already its broad distribution (see below). Abiα also shares similarity with Pfam entry PF10987 and PDB entry 3H35 without providing any substantial information about Abiα function.

To characterize the Abiα mode of action, lysis curves were conducted and indicated that strain pp^−^ containing plasmid pJL3 lyses ~10 min earlier than its isogenic strain devoid of *abiα* after infection by Idefix (MOI 10) (Figure 5A). The same results were obtained when comparing lysis curves of the prophage positive and prophageless strains pp6^+^ and pp^−^, respectively (Figure 5B). We concluded that Abiα provokes a lysis asynchrony.

To evaluate the prevalence of *abiα* and its genetic context, the protein sequence was searched in the JGI database with the IMG interface ([82]; BLASTP, E-value < 3.10^−8^, Identity ≥ 25%). Homologs of Abiα were found mostly on prophages, with some of them shown in Appendix A: the closer relatives (61–59% sequence identity) came from various enterococci, such as *E. faecium* and *Enterococcus villorum* isolates, and subsequent analyses with the NCBI nt database also revealed a homolog in an *Enterococcus hirae* prophage (not shown in Appendix A). Other Abiα were found in about 50 *Lactobacillus* strains (27–42% sequence identity). Roughly, half of these genes are encoded on prophages (as in *L. johnsonii*, *L. salivarius* or *L. plantarum* isolates) whereas the other half are located on the chromosomes (as in *L. lindneri* and *L. helveticus* isolates). Interestingly, in *L. helveticus*, and in two *Oenococcus* species (*O. kitaharae* and *O. oeni)*, the *abiα* gene is close to other putative Abi systems, and may belong to a phage resistance island. Sporadic occurrences (31–37% sequence identity) were also found in three *Streptococcus* species (*S. suis*, *S. bovis*, and *S. parasanguinis*), and one *Carnobacterium* isolate. A prophage context was detectable in this latter and in *S. parasanguinis.* Additional prophage-encoded genes with 25–27% sequence identity with *abiα* were located in one *Lactococcus lactis* ssp. *lactis* (not shown in Appendix A), *Bacillus velezensis, Staphylococcus epidermidis*, *Staphylococcus scuiri* and *Staphylococcus aureus*. We conclude that Abiα is widely distributed, and hypothesize it has a phage origin.

#### 3.2.2. Mutagenesis in V583 *epa* Variable Region as Potential Additional Line of Defense

During this work, we regularly observed phage-resistant colonies growing within lysis zones of the pp6^−^ strain. Fluctuation tests of Luria-Delbrück allowed estimating that the frequency of mutations resulting in resistance of the strain was 1.56 (± 0.54) × 10^−4^ per cellular division. We conclude that spontaneous resistance to Idefix arises at high frequency in *E. faecalis* strain devoid of *abiα.*


To gain further insights into the bacterial mechanisms of resistance to Idefix, we tested nine spontaneous Idefix resistant mutants randomly isolated from fluctuation tests. Idefix adsorption was reduced for all mutants: three of them had a 2-to-4-fold reduced adsorption efficiency and the remaining six, rather a 10-fold defect (Appendix A). We conclude that bacterial resistance to Idefix is due to an adsorption defect.

More than 10 years ago, a study of the podophage C1 distantly related to Idefix (see Figure 2) had proposed that the *N*-acetylgalactosamine (GalNac) composing the side chains of the group C streptococci surface rhamnopolysaccharide should be a crucial element of the bacterial receptor [83]. Enterococci, like group C streptococci, harbor a surface rhamnopolysaccharide or Epa (Enterococcal polysaccharide antigen), which is synthesized by more than thirty proteins encoded within a single cluster of genes. The upstream part of this gene cluster is shared among *E. faecalis* strains, while the immediately downstream region is variable between strains [84]. Within this *epa* variable region, *epaX* encodes a glycosyltransferase involved in the incorporation of galactose and/or GalNac proposed to form the side chains of the rhamnopolysaccharide in *E. faecalis* [49]. We therefore started by examining the *epaX* region in the nine mutants. PCR amplification of *epaX* led to a DNA product of increased size in six out of nine mutants (~4000 instead of ~2500 bp). Sequencing revealed that *epaX* gene was disrupted by an IS256 insertion sequence, positioned in either direction, and at six different locations, all of them in the very distal 3’ region of *epaX* (Figure 6 and Appendix A). 

We next reconstructed a deletion of *epaX* in the pp6^−^ background. Again, Idefix was almost unable to adsorb to the resulting strain (VE18393, Appendix A) with only 5% (± 8) of adsorption, nor to lyse it (Figure 7A). Complementation of this *epaX* mutant with plasmid pVE14176 expressing constitutively *epaX* under a strong promoter (strain VE18945, Appendix A) restored Idefix adsorption, which reached 96% (± 2), as well as the capacity of plaquing (Figure 7B). Results on the isogenic strain (VE18944, Appendix A) hosting the empty vector were 5% (± 6) of phage adsorption and no plaque formation (Figure 7C). We concluded that *epaX* is required for Idefix adsorption. 

To characterize the three remaining Idefix-resistant mutants, in which no IS was detected in the *epaX* region by PCR, we sequenced the genome of one of them, and mapped the reads against the pp6^−^ reference genome. We identified a 33 nt-long deletion, flanked by 11 nt-long direct repeats, within ORF *ef2169* encoding a predicted O antigen polymerase, and located immediately downstream of *epaX*. PCR amplification and sequencing of *ef2169* gene in the two last mutants allowed to detect the same 33 nt deletion in both of them (Figure 6 and Appendix A). The observed deletion results in an in-frame 11 amino acids deletion in the C-terminal region of the putative O antigen polymerase. Additional single nucleotide polymorphisms (introducing amino acid substitution and a premature stop codon, respectively) were also observed in these two last *ef2169* mutants (Appendix A). Complementation of the *ef2169* mutation could not be checked, due to our failure to clone the *ef2169* gene on a plasmid, despite repeated attempts in *E coli*, *L. lactis*, and *E. faecalis*.

## 4. Discussion

Enterococci are particularly successful at rapidly acquiring resistance to virtually any antibiotic used in therapy, with vancomycin-resistant enterococci (VRE) being a major clinical problem. If phage therapy is consequently nowadays (re)considered as an alternative to combat VRE infections, we are still nowhere near using enterococci phages as therapeutic agents in routine. Before that, we need to better characterize (i) enterococci phage receptors that define phage strain specificity, and (ii) enterococci resistance mechanisms that constitute the major constraint of phage therapy [7,16]. 

The present study with the new virulent podophage, Idefix, fulfills these goals. The phage was isolated by using the reference vancomycin-resistant *E. faecalis* V583 clinical isolate, deleted from its endogenous plasmids and prophages, as a recipient for phage infections. Idefix belongs to the *Picovirinae* subfamily and P68 genus and is closely related to three phages recently isolated on *E. faecalis* [13,44] or *E. faecium* [14]. They encode a DNA polymerase from the B type superfamily, which is a hallmark of the *Picovirinae*. As an additional hallmark of these enterococci podophages, they all encode two types of predicted endolysins. Endolysin engineering has emerged as a suitable strategy for food safety, environment decontamination, and infections control [85]. Some of the characterized enterococci phage endolysins show potential to combat VRE in vivo [15,86] and could represent an alternative to resolve the VRE problem. Idefix both encodes endolysin IDF_15, a predicted N-acetylmuramoyl-L-alanine amidase for which we could presume activity by spot assays on bacterial lawns, and IDF_13, similar to the catalytic subunit of the endolysin PlyC encoded by *Streptococcus* phage C1, which was inactive in the same spot assay. PlyC is the unique example of multimeric endolysin and represents the more active peptidoglycan hydrolase reported to date [85]. We were unable to identify by similarity an Idefix gene that could encode the CBD of this complex, and allow the formation of a putative multimeric endolysin. CBD of endolysins bind to specific substrates on the bacterial surfaces, often giving rise to near-species-specific binding, so that endolysins of the same class often share very little sequence similarity in the binding region [85]. Identification of the CBD subunit homolog in Idefix and complete characterization of this intriguing endolysin requires further investigation.

Idefix was unable to lyse any of our fifty-nine strains of *E. faecalis* and *E. faecium*, suggesting it could originate from another enterococcal species. Plaques were obtained against the indicator strain used for its isolation, and not on the wild type isolate V583 from which the indicator strain is derived. This apparently narrow host range could be explained by the reported Abiα-sensitivity and perhaps more importantly by the receptor-specificity of Idefix. 

In V583, resistance to the phage was mainly due to the gene *abiα* (formerly ef2833) encoded by prophage 6. Upon Idefix infection, cells are dying while the Idefix lytic cycle is perturbed (ECOI of ~40% and ~10% whenever *abiα* is expressed on prophage 6 or overexpressed on cloning vectors, respectively) so that its efficiency of plaquing is below 10^−8^ and the detectable limits of the assay for Idefix. To our knowledge, it is the first time that an Abi system is described within enterococci, and the first one targeting a *Picovirinae*. This new Abi protein is widely distributed, it corresponds to the PFAM DUF 4393, and we report here that it is mostly found in prophages, arguing that it is essentially a temperate phage weapon to fight against other phages. Abi are specific anti-phage defence mechanisms present in many bacterial species including *Shigella dysenteriae*, *Streptococcus pyogenes*, *Vibrio cholerae*, *Bacillus subtilis*, and *Bacillus licheniformis* [62,87,88,89,90]. They have been more extensively studied in *L. lactis* and *E. coli* [17,81]. Most *abi* genes are encoded by a single gene on mobile genetic elements (MGE), on prophages in *E coli* [91], or plasmids in *L. lactis* [81]. The fact that most *abi* genes are plasmidic in *L. lactis* may result however from a bias of human selection for plasmid-encoded systems in the dairy industry. Abi actually include a large collection of diverse mechanisms acting at any stage of the phage development to decrease or completely block virion production and cause host cell death. All these mechanisms display little or no known evolutionary relationship, apart from a very similar phenotype [18,81,92].

Lysis curve experiments show that the expression of *abiα* triggers a premature lysis of Idefix-infected *E. faecalis*. A similar phenotype was described in *L. lactis*, in which lysis, upon infection by siphophages from the P335 group, occurs earlier in the presence of AbiZ [93]. AbiZ speeds up the lysis of *L. lactis* bacteria in which the phage holin and endolysin are expressed, and enhances membrane permeability of bacteria expressing the phage holin. Durmaz and Klaenhammer propose that AbiZ interacts with phage holin to accelerate the lysis clock and prevent normal phage multiplication [93]. Abiα and AbiZ do not share any sequence similarity. We were unable to obtain Idefix mutants that were resistant to Abiα, suggesting a very high efficiency of this Abi system. Based on our results, we can only speculate that Abiα interferes with lysis, possibly by preventing timely holin and consequently endolysin(s) actions, as described for AbiZ. Another possibility is that Abiα targets a putative Idefix holin inhibitor. The holin triggering, which is determinant for optimal burst size, is critically regulated by the expressed ratio between the holin and its inhibitor. This latter is encoded within the lambda holin gene, as a separate transcript initiated at a dual-start motif [94,95]. Unlike lambda (and phi29), the Idefix holin gene does not harbor a dual-start motif and its regulation may be mediated by the expression of another non-identified Idefix gene, as described in other phages [94,95]. Finally, premature triggering of lambda holin can also be induced by any poison that efficiently reduces the proton-motrice force (pmf) of the cytoplasmic membrane [94,95]. Thus, we cannot exclude Abiα causes a holin-independent reduction of the pmf underlying an early trigger of the holin and/or a permeabilization of the membrane leading to endolysin release. If this is the case, you might imagine that Abiα could be active against a wider range of phages and not only Idefix or related phages. Further experiments will be needed to test these different hypotheses and complete the characterization of this defence mechanism. 

Genomes of V583 and other MDR *E. faecalis* isolates have few generalist anti-MGE systems, which favor horizontal gene transfer and polylysogeny [29]. MDR enterococci may thus provide a suitable ground for bacteriophages confrontation. Whereas the spread of Idefix infection is thus limited by V583 prophage 6, it is interesting to note this latter is itself strictly controlled by V583 prophages 3 and 5 that block its excision [43]. This ‘domestication’ could be seen as a way for both prophages to sustain the specific line of defense conferred by prophage 6 and thus protect themselves and their host from an external phage ‘attack’.

Beyond the intra-bacteriophage warfare, bacteria are also able to evolve resistance to phages. We observed that *E. faecalis* mutants resisting to Idefix infection arose at a high frequency of 1.5 × 10^−4^ per generation. The nine Idefix-resistant mutants analyzed had either acquired an IS256 or recombined between eleven base paires long direct repeats in the *epa* variable region, indicating that *E. faecalis* has a high potential for evolution by recombination. Mutants in glycosyltransferase gene *epaX* had a 10-fold defect in Idefix adsorption, whereas mutants in the next o-polymerase gene had a milder adsorption defect. The fact that all mutations cluster into two genes of the *epa* variable region underlines the role of the rhamnopolysaccharide for Idefix adsorption. Epa was recently confirmed as a receptor of *E. faecalis* virulent phage NPV1 [96,97], and we provide here the first evidence for a role of Epa decoration chains in phage/host recognition in enterococci. Rhamnose-rich cell wall polysaccharides (CWPS) and their structural diversity are important in phage adsorption within lactococci and streptococci [98,99]. In *L. lactis* as well, CWPS biosynthesis is encoded on a large chromosomal gene cluster also including a conserved and a variable region. Based on sequence similarity and difference in the variable region, *L. lactis* strains were divided into three groups, and one of them in several subtypes [100]. These latter are distinguished by their glycosyltransferase-encoding genes composition and consequent differences between CWPS structures are critical in determining phage sensitivity [100]. The use of unconserved sugar decorations of the saccharidic chains as a receptor combined with the presence of a prophage-encoded Abi system targeting Idefix may explain Idefix very low success in *E. faecalis*. 

## 5. Conclusions

This study of the *Enterococcus* infecting *Picovirinae* Idefix permitted to unveil two resistance mechanisms against it, one bacterial and the other viral. Bacterial mutations suppressed the Epa decoration needed for phage adsorption, and the prophage-encoded product Abiα interfered with Idefix timing of lysis. However, the bacterial line of defence is likely to be counterselected in vivo as V583 *epaX* mutants have a defect in mouse gut colonization [49]. In fact, phage selective pressure might be one of the actors of the observed diversification of the *epa* locus in enterococci. The viral line of defence based on Abiα is likely to be more robust, given that no escape mutant could be isolated. Indeed, Abiα is widespread both in enterococci and in *Firmicutes* and it may allow to fight against *Picovirinae* or other ranges of phages. Somehow, the different families of phages infecting the same bacterial species are in competition, so that their host resembles a battlefield. It is especially true if bacteria are devoid of generalist defense systems, as MDR *E. faecalis* strains tend to be. In this context, it might be expected that Abi and other specific defensive systems pave prophage genomes, allowing temperate phages to align their strategy with their host, and fight against virulent phage invaders.

## Figures and Tables

**Figure 1 viruses-11-00048-f001:**
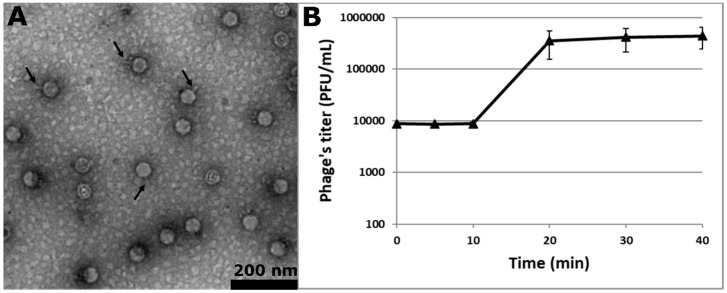
Characterization of the virulent phage Idefix. (**A**) Electron micrograph of Idefix particles negatively stained with 2% uranyl acetate (some Idefix tails are shown by arrows). (**B**) One step growth kinetic of phage Idefix determined in its host strain *E. faecalis* pp^−^. The values indicate the means and standard deviations of three independent experiments.

**Figure 2 viruses-11-00048-f002:**
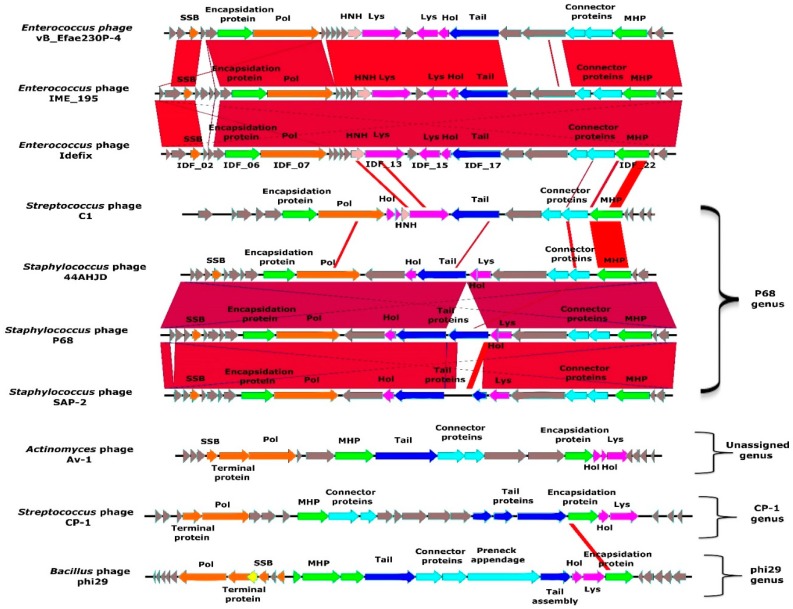
Annotation and comparison of the phage Idefix genome. *Enterococcus* phage genomes from vB_Efae230P-4, vB_EfaP_IME195, and Idefix compared to *Streptococcus* phage C1, *Staphylococcus* phages 44AHJD, P68, SAP-2, 66, *Actinomyces* phage Av-1, *Streptococcus* phage CP-1 and *Bacillus* phage phi29 all belonging to the *Picovirinae* subfamily. Gene functions are color-coded and detailed (yellow: transcriptional regulation, orange: DNA metabolism, green: DNA packaging and head, light blue: head to tail, dark blue: tail, pink: HNH endonuclease, fuchsia: lysis, grey: hypothetical proteins).

**Figure 3 viruses-11-00048-f003:**
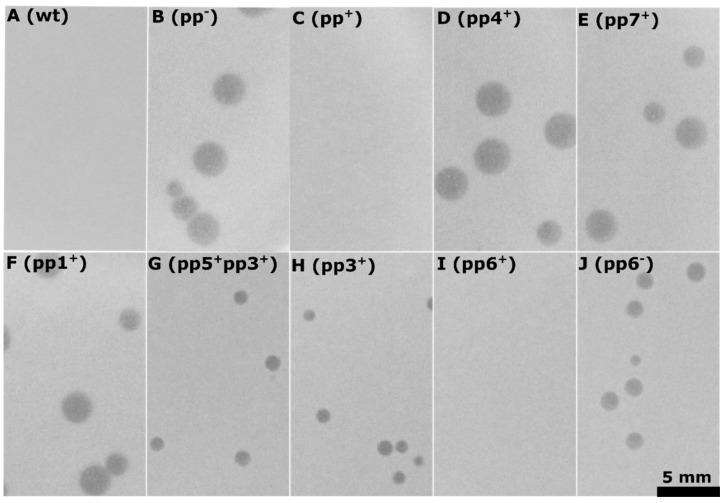
V583 endogenous prophages (pps) interference with Idefix infection, plaque assays performed on strains: A. wt (VE14002), V583 clinical isolate. B. pp- (VE18590), a V583 derivative deleted for all its plasmids and prophages. C. pp^+^ (VE14089), a V583 derivative only deleted from all its plasmids. D. pp4^+^ (VE18582), a VE14089 derivative deleted for all its pps but pp4. E. pp7^+^ (VE18589), a VE14089 derivative deleted for all its pps but pp7. F. pp1^+^ (VE18562), a VE14089 derivative deleted for all its pps but pp1. G. pp3^+^pp5^+^ (VE18583), a VE14089 derivative deleted for all its pps but pp3 and pp5. H. pp3^+^ (VE18584), a VE14089 derivative deleted for all its pps but pp3. I. pp6^+^ (VE18581), a VE14089 derivative deleted for all its pps but pp6. J. pp6^−^ (VE18306), a VE14089 derivative only deleted from pp6.

**Figure 4 viruses-11-00048-f004:**
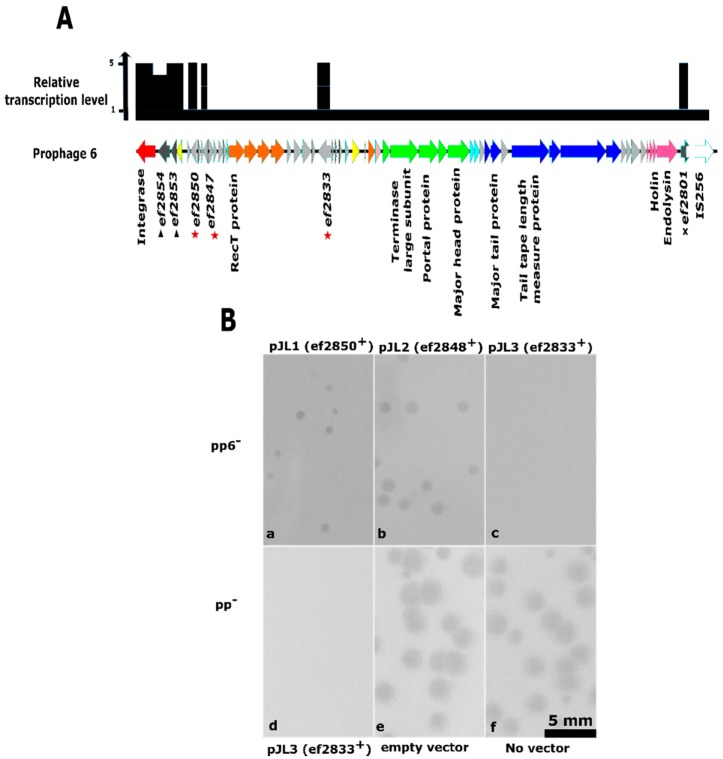
Identification of the V583 endogenous prophage 6 (pp6) gene responsible for Idefix resistance. (**A**) Genomic organization of pp6 and relative transcription level of each gene. The genomic organization of pp6 is modified from [43]. Gene functions are color-coded and detailed wherever possible (red: integrase, yellow: transcriptional regulation, orange: DNA metabolism, green: DNA packaging and head, light blue: head to tail, dark blue: tail, fuchsia: lysis, grey: hypothetical proteins, white: transposable element). Relative transcriptional level of each gene is indicated in black and designed from a previously published transcriptomic study [48]. Eight pp6 genes are constitutively expressed during V583 normal growth conditions. Among those not involved in pp6 lysogenic decision: a membrane protein and putative metallo-peptidase genes are indicated with triangles, a putative glycosyltransferase gene disrupted by a transposable element is indicated with a cross and the three other genes tested as *abi* candidates are indicated with red stars. (**B**) Interference with Idefix infection following the expression of the three pp6 *abi* candidates, plaque assays performed on strain pp6^−^ (A to C) or pp^−^ (D to F) transformed by: **a.** pJL1. **b.** pJL2. **c.** pJL3. **d.** pJL3. **e.** the pJIM2246 empty vector. **f.** no vector.

**Figure 5 viruses-11-00048-f005:**
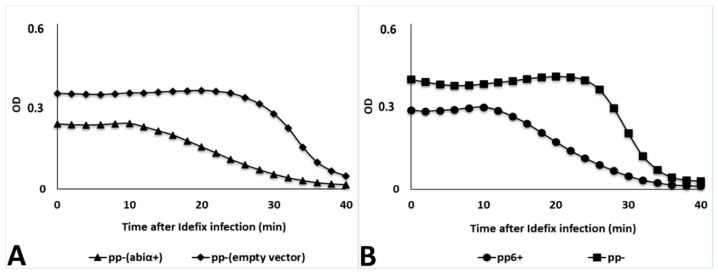
Lysis curves after Idefix infection at MOI 10. (**A**) Comparison between pp^−^pJIM2246*abiα*^+^ (VEJL5) and pp^−^ pJIM2246 (VEJL4) in triangles and diamonds, respectively. (**B**) Comparison between pp6^+^ and pp^−^ in circles and squares, respectively. The results shown are representative of three biological replicates (see Appendix A).

**Figure 6 viruses-11-00048-f006:**
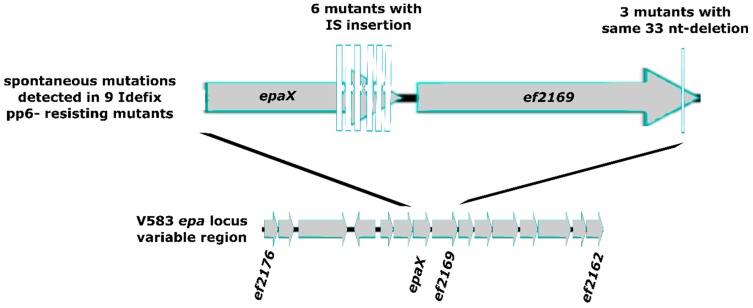
Genomic organization of V583 variable *epa* locus and focus on spontaneous mutations found in *epaX* region for nine Idefix pp6^−^ resisting mutants.

**Figure 7 viruses-11-00048-f007:**
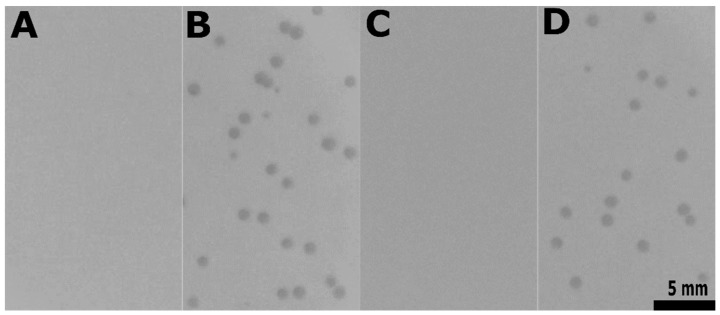
*epaX* gene expression is required for Idefix infection of pp6^−^. Plaque assays performed on strains: (**A**) pp6^−^*epaX*^−^ (VE18393), a VE18306 derivative deleted from *epaX*. (**B**)pp6^−^*epaX*^−^pVE14176*epaX*^+^ (VE18945), a VE18393 derivative complemented with *epaX*. (**C**) pp6^−^*epaX*^−^pVE14176 (VE18944), a VE18393 derivative carrying the empty vector. (**D**) pp6^−^ (VE18306), a V583 derivative deleted from its plasmids and pp6.

**Table 1 viruses-11-00048-t001:** Efficiency of plaquing (EOP) of phage Idefix on *E. faecalis* V583 derivatives deleted from different prophages.

Strain (Genotype)	Average EOP of Idefix (sd)	Plaque (Diameter)
VE18590 (pp^−^)	1	Big clear plaque (3–4 mm)
VE18562 (pp1^+^)	1.1 (0.17)	Big clear plaque (3–4 mm)
VE18582 (pp4^+^)	0.92 (0.14)	Big clear plaque (3–4 mm)
VE18589 (pp7^+^)	1.03 (0.16)	Big clear plaque (3–4 mm)
VE18583 (pp3+, pp5^+^)	0.97 (0.16)	Reduction in size (1 mm)
VE18584 (pp3^+^)	0.92 (0.17)	Reduction in size (1 mm)
VE18581 (pp6^+^)	0 *	No plaque visible
VE18306 (pp6^−^)	1 (0.13)	Reduction in size (1 mm)

* Below the detectable limit of the assay, at least < 1.2 × 10^−8.^

**Table 2 viruses-11-00048-t002:** Parameters of phage Idefix proliferation on *E. faecalis* strains differing by the presence or absence of prophage 6 or prophage 6-encoded *ef2833.*

Strain (Genotype)	% EOP (Plaques Diameter)	% Adsorption (sd)	% ECOI (sd)	% Bacterial Death at MOI 1 (sd) and MOI 10 (sd)
Indicator strain VE18590 (pp^−^)	100 (3–4 mm)	96.5 (1.63)	100	69.0 (0.39) and 99.2 (0.78)
VE18306 (pp6^−^)	100 (1 mm)	97.1 (1.35)	69.3 (3.10)	52.8 (3.00) and 98.2 (0.98)
VE18581 (pp6^+^)	0 None plaques visible	97.5 (0.92)	39.5 (3.94)	66.8 (2.78) and 98.2 (1.32)
VE14089 (pp^+^)	0 None plaques visible	98.1 (2.36)	15.9 (4.01)	60.2 (3.46) and 98.9 (0.19)
VEJL3 (pp6^−^, *ef2833*^+^)	0 None plaques visible	98.4 (0.68)	9.46 (2.68)	67.7 (1.65) and 98.6 (0.46)
VEJL5 (pp^−^, *ef2833*^+^)	0 None plaques visible	98.5 (1.33)	13.8 (0.58)	63.2 (1.98) and 96.6 (1.36)

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
