# Peer review of "Enterococcus faecalis* Countermeasures Defeat a Virulent *Picovirinae* Bacteriophage"

_viruses, 2019, doi:10.3390/v11010048_

Round 1
Reviewer 1 Report
This manuscript describes the isolation of a novel virulent Enterococcus phage and subsequent identification of two antiphage defence mechanisms respectively encoded in the bacterial chromosome and a prophage. Given the current interest in developing phage therapy as a viable alterantive or supplement to antibiotics. The article is well organized and easy to follow, but it would benefit from English editing. Additional comments: - Line 488: S. scuiri and S. aureus should be changed to Staphylococcus sciuri and Staphylococcus aureus
Author Response
Response: According to Reviewer 1 request, we performed English editing:
-“Enterococcus faecalis is an opportunistic pathogen that has emerged as a major cause of nosocomial infections worldwide.”
(Lines 16-17)
- “We made use of a prophageless derivative of the well-known clinical strain E. faecalis V583 to isolate a virulent phage belonging to the Picovirinae subfamily and to the P68 genus that we named Idefix.”
(Lines 20-22)
-“Apart from this, the modes of defence of enterococci against phages largely remain a terra incognita.”
(Lines 20-22)
-“Keeping in mind that in most ecosystems, bacteria co evolve with a plethora of bacteriophages, which imperatively depend on them for their reproduction, it is no surprise that bacteria invest efforts in fighting against them.”
(Lines 53-55)
Finally, no complete defence island system associated with restriction-modification was detected in E. faecalis to date [22]
(Lines 62-63)
“We shew that E. faecalis prophage 6, which is apparently derived from a siphophage, is especially efficient in resisting to Idefix infection due to a new abortive infection system that we call Abiα.”
(Lines 99-101)
“Transformations of chemically competent E. coli ER2566 were carried out by a heat shock procedure.”
(Line 245-246)
We replaced “curated” by “cured of” in the manuscript.
(Lines 89, 95, 110, 228 and 274)
We replaced “led” by “performed” in the manuscript.
(Lines 187, 206 and 238)
We replaced “esperance” by “expectation” in the manuscript.
(Line 236)
We also changed “S. scuiri and S. aureus” to “Staphylococcus sciuri and Staphylococcus aureus”
(Lines 498-499)
Reviewer 2 Report
This manuscript is packed with new information. It has a new and well characterized phage sequence placed into a useful context with the rest of its subfamily, a new and well characterized abi system, and additional information about phage resistance in E. faecalis. This whole topic of the biology of and interactions with and among resident prophages is in need of contributions of this kind.
I have no major criticisms, but I do have a request of the authors. Once the paper is accepted, please go to the Pfam entry for PF14337 and use the "provide feedback" button to add a summary of your functional information and a citation to that entry.
Minor issues:
From hhpred, ef2833 also has similarity to PF10987, which seems to be a Gram - version of the same family, and to pdb entry 3H35. Neither of those provides any substantial information about function, but I'd encourage mentioning all the available information about this homology group.
From your prior characterization of prophage 6, can you tell that it's clearly not derived from a podovirus?
line 56-65: The compendium of information provided here is important. But, the text seems to imply that phage resistance is usually encoded by core genome components in other bacterial species. Was that unintended, or can you provide citation to such a precedent?
Line 62: "only one gene composing defence island system" There is some grammatical problem here.
Line 95: "V583 derivative strain curated for all plasmids and the six active prophages"; That's not what "curated" means. I think you meant "cured of"? Occurs several other places in text.
line 186: "These experiments were independently led three times" The word "led" is used several times in the text in this way. I think you meant you meant "performed" or "conducted".
line 234: "is related to h, the Esperance of this law"; I'm guessing "esperance" is French for "expectation".
p { margin-bottom: 0.1in; line-height: 120%; }
Author Response
This manuscript is packed with new information. It has a new and well characterized phage sequence placed into a useful context with the rest of its subfamily, a new and well characterized abi system, and additional information about phage resistance in E. faecalis. This whole topic of the biology of and interactions with and among resident prophages is in need of contributions of this kind.
I have no major criticisms, but I do have a request of the authors. Once the paper is accepted, please go to the Pfam entry for PF14337 and use the "provide feedback" button to add a summary of your functional information and a citation to that entry.
Response: We agree with this request and we will reply to it once the paper will be accepted.
From hhpred, ef2833 also has similarity to PF10987, which seems to be a Gram - version of the same family, and to pdb entry 3H35. Neither of those provides any substantial information about function, but I'd encourage mentioning all the available information about this homology group.
Response: We added this information by writing: “Abiα also shares similarity with Pfam entry PF10987 and PDB entry 3H35 without providing any substantial information about Abiα function.”
(Lines 472-473)
From your prior characterization of prophage 6, can you tell that it's clearly not derived from a podovirus?
Response: BlastP analyses reveal that prophage 6 major head protein and major tail protein harbor siphoviral conserved domains from HK97 family and phi13 family respectively. This leads us to believe that prophage 6 is derived from a siphovirus.
We can add this information by writing: “We shew that E. faecalis prophage 6, which is apparently derived from a siphophage, is especially efficient in resisting to Idefix infection due to a new abortive infection system that we call Abiα.”
(Lines 99-101)
And “We concluded that prophage 6, apparently derived from a siphophage [43], is sufficient for Idefix resistance.”
(Lines 397-399)
line 56-65: The compendium of information provided here is important. But, the text seems to imply that phage resistance is usually encoded by core genome components in other bacterial species. Was that unintended, or can you provide citation to such a precedent?
Response: It is indeed almost unintended. We consequently tried to reformulate the paragraph and wrote: “The only common clustered regularly interspaced short palindromic repeat (CRISPR)-CRISPR associated (Cas) locus found in E. faecalis isolates has lost its cas genes, and two complete type II CRISPR-Cas systems occur variably across the species [24–26]. Furthermore, no restriction-modification (R-M) system is commonly found within the species , and to date only a single type II R-M system has been described in three E. faecalis chromosomes [27,28].
(Lines 57-62)
Line 62: "only one gene composing defence island system" There is some grammatical problem here.
Response: We fixed the grammatical problem by writing: “Finally, no complete defence island system associated with restriction-modification was detected in E. faecalis to date.” (Lines 62-63)
Line 95: "V583 derivative strain curated for all plasmids and the six active prophages"; That's not what "curated" means. I think you meant "cured of"? Occurs several other places in text.
Response: We replaced “curated” by “cured of” in the manuscript.
(Lines 89, 95, 110, 228 and 274)
line 186: "These experiments were independently led three times" The word "led" is used several times in the text in this way. I think you meant you "performed" or "conducted".
Response: We replaced “led” by “performed” in the manuscript.
(Lines 187, 206 and 238)
line 234: "is related to h, the Esperance of this law"; I'm guessing "esperance" is French for "expectation".
Response: We replaced “esperance” by “expectation” in the manuscript.
(Line 236)